# What do we know about SARS-CoV-2 transmission? A systematic review and meta-analysis of the secondary attack rate and associated risk factors

Wee Chian Koh[1], Lin Naing[2], Liling Chaw[2], Muhammad Ali Rosledzana[3], Mohammad Fathi Alikhan[3], Sirajul Adli Jamaludin[3], Faezah Amin[3], Asiah Omar[3], Alia Shazli[3], Matthew Griffith[4], Roberta Pastore[4], Justin Wong[3]*

1 Centre for Strategic and Policy Studies, Brunei Darussalam, Bandar Seri Begawan, Brunei, 2 PAPRSB Institute of Health Sciences, Universiti Brunei Darussalam, Bandar Seri Begawan, Brunei, 3 Disease Control Division, Ministry of Health, Brunei Darussalam, Bandar Seri Begawan, Brunei, 4 Western Pacific Regional Office (Manila), World Health Organization, Manila, Philippines

* justin.wong@moh.gov.bn

**Data Availability Statement:** All relevant data are within the manuscript and its Supporting Information files.

## Abstract

### Introduction

Current SARS-CoV-2 containment measures rely on controlling viral transmission. Effective prioritization can be determined by understanding SARS-CoV-2 transmission dynamics. We conducted a systematic review and meta-analyses of the secondary attack rate (SAR) in household and healthcare settings. We also examined whether household transmission differed by symptom status of index case, adult and children, and relationship to index case.

### Methods

We searched PubMed, medRxiv, and bioRxiv databases between January 1 and July 25, 2020. High-quality studies presenting original data for calculating point estimates and 95% confidence intervals (CI) were included. Random effects models were constructed to pool SAR in household and healthcare settings. Publication bias was assessed by funnel plots and Egger's meta-regression test.

### Results

43 studies met the inclusion criteria for household SAR, 18 for healthcare SAR, and 17 for other settings. The pooled household SAR was 18.1% (95% CI: 15.7%, 20.6%), with significant heterogeneity across studies ranging from 3.9% to 54.9%. SAR of symptomatic index cases was higher than asymptomatic cases (RR: 3.23; 95% CI: 1.46, 7.14). Adults showed higher susceptibility to infection than children (RR: 1.71; 95% CI: 1.35, 2.17). Spouses of index cases were more likely to be infected compared to other household contacts (RR: 2.39; 95% CI: 1.79, 3.19). In healthcare settings, SAR was estimated at 0.7% (95% CI: 0.4%, 1.0%).

**Funding:** The author(s) received no specific funding for this work.

**Competing interests:** The authors have declared that no competing interests exist.

## Discussion

While aggressive contact tracing strategies may be appropriate early in an outbreak, as it progresses, measures should transition to account for setting-specific transmission risk. Quarantine may need to cover entire communities while tracing shifts to identifying transmission hotspots and vulnerable populations. Where possible, confirmed cases should be isolated away from the household.

## Introduction

The COVID-19 pandemic continues to escalate. Modeling studies have enhanced understanding of SARS-CoV-2 transmission dynamics and initial phylogenetic analysis of closely related viruses suggest highly linked person-to-person spread of SARS-CoV-2 originating from mid-November to early December 2019 [1–3].

There are no known effective therapeutics or vaccines [4, 5]. As such, containment measures rely on the capacity to control viral transmission from person-to-person, such as case isolation, contact tracing and quarantine, and physical distancing [6]. Effective prioritization of these measures can be determined by understanding SARS-CoV-2 transmission patterns.

There is an abundance of literature on the biological mode of transmission of coronaviruses: through exhaled droplets, aerosol at close proximity, fomites, and possibly through fecal-oral contamination [7, 8]. However, few observational studies have assessed transmission patterns in populations, and what determines whether the infection is contained or spreads. Previous theoretical work by Fraser et al. proposed three transmission-related criteria that impact on outbreak control: (i) viral transmissibility; (ii) disease generation time; and (iii) the proportion of transmission occurring prior to symptoms [9].

To better understand SARS-CoV-2 transmission, we conducted a systematic review and meta-analyses of publicly available studies to estimate the secondary attack rate (SAR) in various settings. We also examined whether household transmission differed by symptom status of index case, adult and children (< 18 years old), and relationship to index case.

## Methods

This systematic review and meta-analysis followed the Preferred Reporting Items for Systematic Reviews and Meta-Analyses (PRISMA) guidelines.

### Definition

SAR is defined as the probability that an exposed susceptible person develops disease caused by an infected person [10]. It is calculated by dividing the number of exposed close contacts who tested positive (numerator) by the total number of exposed close contacts of the index case (denominator).

### Search strategy and inclusion criteria

We performed a literature search of published journal articles in PubMed and pre-print articles in medRxiv and bioRxiv from January 1, 2020 using the search terms ("SARS-CoV-2" OR "COVID-19") AND ("attack rate" OR "contact tracing" OR "close contacts"). The last search date was on July 25, 2020. All studies that were written in English or have an abstract in English were included.

Studies reporting SAR were included if they: (i) presented original data for SAR estimation, such as from a contact tracing investigation; (ii) reported a numerator and denominator of close contacts, or at least two of numerator, denominator, and SAR; (iii) specified a particular setting; and (iv) cases were confirmed positive with SARS-CoV-2 through reverse transcription polymerase chain reaction (RT-PCR) test. Point-testing or prevalence studies to measure cumulative incidence of infection in a setting were excluded from the meta-analyses as the source of infection could not be traced, but we discussed some of these studies where relevant.

### Data extraction and quality assessment

The articles were initially screened by title and abstract, and subsequently by review of selected full-text articles. Three reviewers selected the studies independently using predetermined inclusion criteria and differences in opinions were resolved through consensus. Data were obtained directly from the reports, but when not explicitly stated, we derived the data from tables, charts, or supplementary materials. The following data were extracted from each included study: surname of first author; study design; location of study; number of index cases; total number of close contacts; number of close contacts tested positive for SARS-CoV-2; setting type; symptom of index case; age group of secondary cases; and relationship of secondary cases to index case.

The quality of the studies was independently assessed by three reviewers based on the UK National Institute for Clinical Excellence guidelines [11]. The evaluation is based on a set of eight criteria. Differences in assessments were resolved through consensus. Studies with a score greater than 4 (out of 8) were considered to be of high quality and thus included in the meta-analyses [12].

### Statistical analysis

Point estimates and 95% confidence intervals (CI) were calculated. CIs were estimated using a Normal approximation but in studies with a small number of secondary cases ($< 5$) a binomial approximation was used. Meta-analyses were performed using random-effects DerSimonian-Laird model [13]. We also estimated risk ratios to examine SAR differences by symptom status of the index case, age of close contacts, and relationship of household contacts. The $I^2$ statistic was used as a measure of heterogeneity, with higher values signifying greater degree of variation [14]. Publication bias was assessed by funnel plots and Egger's meta-regression test [15]. A p-value of $<0.05$ was considered as statistically significant. Statistical analysis was done in STATA 14 using the package metan, metafunnel, and metabias [16–18].

## Results

A total of 663 records were identified from the databases (Fig 1). After screening by title and abstract, we included 118 studies and after a detailed assessment based on the inclusion criteria and quality assessment, 57 studies were included in the meta-analyses. A majority of the included studies focused on transmission in households. In non-household settings, most studies were conducted in healthcare settings. As such, our systematic review and meta-analyses focused on SAR in household and healthcare settings, but we also discussed the SAR in other settings.

### Household SAR

We identified 43 studies that allowed direct estimation of the SAR in households (Table 1). Thirty-five studies were published articles (five in Chinese language, two in Korean language)

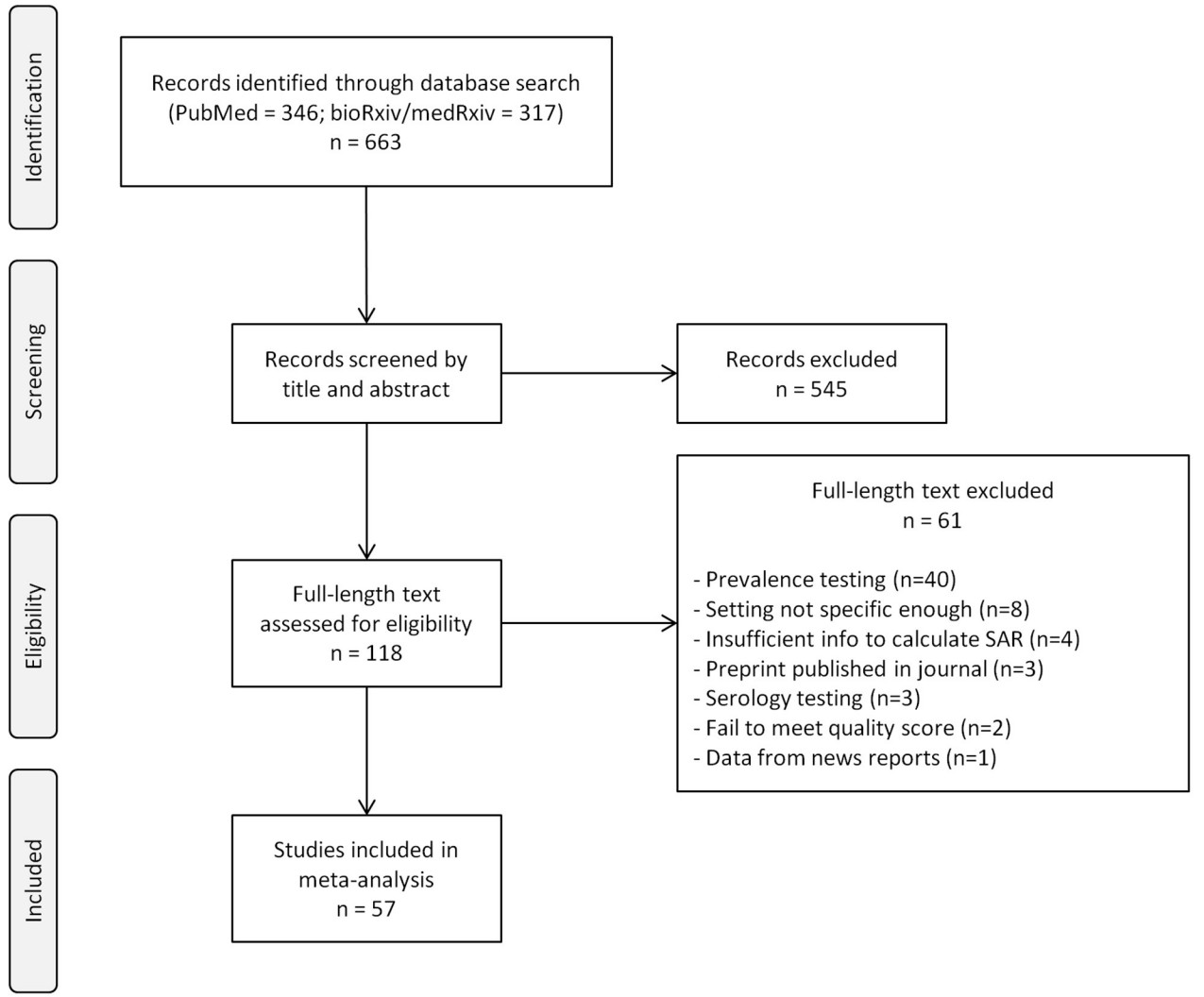

**Fig 1. Flow chart of search strategy and study selection for the secondary attack rate (SAR).**

and eight were pre-prints. About half of the studies were in China (22 in mainland China, 1 in Hong Kong, 1 in Taiwan), five in South Korea, four in the United States, two in Israel, and the others were in Australia, Brunei, Canada, Germany, India, Italy, Singapore, and Spain.

Index cases were confirmed positive cases identified or suspected to have been first exposed to the SARS-CoV-2 virus within the household, generally based on the timing of symptom onset and epidemiological link. Some studies identified close contacts through active surveillance systems while in others they were identified following an outbreak investigation. Testing protocols of close contacts also differed; all close contacts were tested regardless of symptoms in most studies, but only symptomatic contacts were tested in five studies.

There was variation in the definition of household contacts; most included only those who resided with the index case, some studies expanded this to include others who spent at least a night in the same residence or a specified duration of at least 24 hours of living together, while others included family members or close relatives.

**Table 1. Description of studies included in the review and analysis of household secondary attack rate (SAR).**

| Study, location | Description of study | Definition of close contact | Household SAR (%) | No. of index cases | Additional comments | Quality score |
|---|---|---|---|---|---|---|
| Wang et al., Beijing, China [19] | Retrospective study of households | Lived with primary case in a house for 4 days before and for more than 24 hours after the primary case developed illness related to COVID-19 | 77/335 (23.0%) | 41 | | 6 |
| Wang et al., Beijing, China [20] | Summary of contact investigations | Family members or relatives | 111/714 (15.5%) | 585 | | 7 |
| Liu et al., Guangdong, China [21] | Retrospective study of different exposure settings | Spouse and family members | 330/2441 (13.5%) | 1361 | | 7 |
| Jing et al., Guangzhou, China [22] | Retrospective study of households | Lived in the same residential address | 93/542 (17.2%) | 215 | | 6 |
| Luo et al., Guangzhou, China [23] | Prospective study of different modes of contact | Lived in the same household | 96/946 (10.1%) | 347 | | 7 |
| Zhang et al., Guangzhou, China [24] | Retrospective study of pre-symptomatic transmission in different exposure settings | Lived in the same household | 10/62 (16.1%) | 38 | | 6 |
| Wu et al., Hangzhou, China [25] | Retrospective study of different exposure settings | Lived in the same household | 50/280 (17.9%) | 144 | | 5 |
| Li et al., Hubei, China [26] | Retrospective study of households | Lived in the same residence for at least 24 hours | 64/392 (16.3%) | 105 | | 6 |
| Zhang et al., Hunan, China [27] | Retrospective study of different exposure settings | Lived in the same household | 339/617 (54.9%) | 136 | | 6 |
| Zhang et al., Liaocheng, China [28] | Retrospective study of a supermarket cluster | Family members | 12/93 (12.9%) | 25 | | 5 |
| Deng et al., Nanchang, China [29] | Retrospective study of different exposure settings | Lived in the same household | 20/201 (10.0%) | 27 | | 5 |
| Chen et al., Ningbo, China [30] | Prospective study of different exposure settings | Lived in the same household | 37/279 (13.3%) | 187 | | 6 |
| Xin et al., Qingdao, China [31] | Prospective study of households | Family members in the same house | 19/106 (17.9%) | 31 | | 7 |
| Bi et al., Shenzhen, China [32] | Retrospective study of cases identified through symptomatic surveillance and contact tracing | Shared a room, apartment, or other sleeping arrangement | 77/686 (11.2%) | 391 | | 6 |
| Wei et al., Shenzhen, China [33] | Retrospective study of households | Lived in the same household, including visiting period | 21/66 (31.8%) | 60 | | 5 |
| Dong et al., Tianjin, China [34] | Retrospective study of households | Family members | 53/259 (20.5%) | 135 | | 5 |
| Wang et al., Wuhan, China [35] | Retrospective study of household transmission by healthcare workers | Family members | 10/43 (23.3%) | 25 | | 5 |
| Wang et al., Wuhan, China [36] | Retrospective study of households | Lived in the same household | 47/155 (30.3%) | 85 | Only close contacts with symptoms tested; 51 contacts without symptoms assumed negative | 5 |
| Yu et al., Wuhan, China [37] | Retrospective study of different exposure settings | Family members | 143/1396 (10.2%) | 560 | | 5 |
| Hua et al., Zhejiang, China [38] | Retrospective study of households | Family members | 151/835 (18.1%) | n/a | | 7 |
| Sun et al., Zhejiang, China [39] | Retrospective study of family clusters | Family members | 189/598 (31.6%) | 148 | | 5 |
| Wu et al., Zhuhai, China [40] | Retrospective study of households | Spent at least one night in the house after symptom onset of the index case | 48/148 (32.4%) | 35 | | 6 |

(*Continued*)

**Table 1.** (*Continued*)

| Study, location | Description of study | Definition of close contact | Household SAR (%) | No. of index cases | Additional comments | Quality score |
|---|---|---|---|---|---|---|
| Kwok et al., Hong Kong, China [41] | Retrospective study of cases and close contacts | Provided care or stayed at the same place while the index case was ill | 24/206 (11.7%) | 53 | | 6 |
| Cheng et al., Taiwan, China [42] | Prospective study of different exposure settings and different exposure time windows | Lived in the same household | 10/151 (6.6%) | 100 | Only close contacts with symptoms tested | 7 |
| Draper et al., Northern Territory, Australia [43] | Retrospective study in different exposure settings | Lived in the same household | 2/51 (3.9%) | 28 | Only close contacts with symptoms tested | 6 |
| Chaw et al., Brunei [44] | Retrospective study in different exposure settings | Lived in the same household | 28/264 (10.6%) | 19 | | 5 |
| Schwartz et al., Ontario, Canada [45] | Retrospective study of household transmission by healthcare workers | Lived in the same residential address | 391/3986 (9.8%) | n/a | | 6 |
| Böhmer et al., Bavaria, Germany [46] | Analysis of contact investigation | Shared living space | 2/20 (10%) | 1 | | 6 |
| Laxminarayan et al., Tamil Nadu, India [47] | Retrospective study of different exposure settings | Lived in the same household | 380/4066 (9.3%) | 997 | | 7 |
| Boscolo-Rizzo et al., Treviso, Italy [48] | Retrospective study of adult household contacts of mildly symptomatic cases | Lived in the same household | 54/121 (44.6%) | 179 | Only 121 out of 296 close contacts tested | 5 |
| Dattner et al., Bnei Brak, Israel [49] | Summary of contact investigations | Lived in the same household | 981/2824 (34.7%) | 529 | | 6 |
| Somekh et al., Bnei Brak, Israel [50] | Analysis of contact investigation | Lived in the same household | 36/94 (38.3%) | n/a | | 5 |
| Yung et al., Singapore [51] | Retrospective study of paediatric household contacts | Lived in the same household | 13/213 (6.1%) | 223 | | 6 |
| Lee et al., Busan, South Korea [52] | Analysis of contact investigation of asymptomatic index cases | Lived in the same household | 1/23 (4.3%) | 10 | | 5 |
| Son et al., Busan, South Korea [53] | Summary of contact investigations | Lived in the same household | 16/196 (8.2%) | 108 | | 6 |
| Park et al., Seoul, South Korea [54] | Retrospective study of a call center cluster | Lived in the same household | 34/225 (15.1%) | 97 | | 6 |
| Korea CDC, South Korea [55] | Summary of contact investigations | Lived in the same household | 9/119 (7.6%) | 30 | | 5 |
| Park et al., South Korea [56] | Summary of contact investigations | Lived in the same household | 1248/10592 (11.8%) | 5706 | | 7 |
| Arnedo-Pena et al., Castellon, Spain [57] | Retrospective study of households | Lived in the same household | 83/745 (11.1%) | 347 | | 6 |
| Rosenberg et al., New York State, United States [58] | Retrospective study of different exposure settings | Lived in the same residential address | 131/343 (38.2%) | 229 | | 6 |
| Dawson et al., Wisconsin, United States [59] | Retrospective study of households | Lived in the same household | 16/64 (25%) | 26 | | 5 |
| Yousaf et al., Wisconsin and Utah, United States [60] | Retrospective study of households | Lived in the same household | 47/195 (24.1%) | n/a | | 6 |

(*Continued*)

**Table 1.** (Continued)

| Study, location | Description of study | Definition of close contact | Household SAR (%) | No. of index cases | Additional comments | Quality score |
|---|---|---|---|---|---|---|
| Burke et al., United States [61] | Analysis of contact investigation | Family members or friends who spent at least one night in the same residence during the presumed infectious period of the index case | 2/15 (13.3%) | 9 | Only close contacts with symptoms tested | 6 |

Note: Index cases as defined in the respective study, generally determined based on the timing of symptom onset and epidemiological link.

Only three studies differentiated the symptom status of index cases into pre-symptomatic and symptomatic. Fourteen studies had information on age groups that allowed differentiation by children and adults. Seven studies reported SAR by the relationship of close contacts of index cases.

From these 43 studies, we estimated household SAR and conducted subgroup analyses by stratifying according to location, definition of close contact, testing protocol, and publication status. We also examined whether SAR differed by symptom status of index case, child/adult infection, and relationship of close contacts of index cases.

Fig 2 summarizes the estimated SARs. The pooled household SAR is 18.1% (95% CI: 15.7%, 20.6%) with significant heterogeneity (p <0.001). Household SAR ranged from 3.9% in Australia (Northern Territory) to more than 30% in some studies in China (Hunan, Shenzhen, Wuhan, Zhejiang, Zhuhai), Israel (Bnei Brak), Italy (Treviso), and the United States (New York).

## Stratified household SAR

The household SAR from studies in mainland China (20.1%; 95% CI: 16.2%, 23.9%) was not significantly higher than other countries and areas (16.0%; 95% CI: 12.6%, 19.5%) (S1 Fig in S1 Materials). There was no significant difference in SAR in terms of the definition of household close contacts, whether they were based on living in the same household (18.2%; 95% CI: 15.3%, 21.2%) or based on relationships such as family and close relatives (17.8%; 95% CI: 13.8%, 21.8%) (S2 Fig in S1 Materials). Difference in testing protocols—whether testing was done for all contacts regardless of symptoms (18.0%; 95% CI: 15.4%, 20.5%) or symptomatic contacts only (19.8%; 95% CI: 4.6%, 35.0%)—also did not show a significant difference in household SAR (S3 Fig in S1 Materials).

The household SAR for published studies (18.7%; 95% CI: 16.0%, 21.4%) was not significantly higher than preprints (15.6%; 95% CI: 8.7%, 22.4%) (S4 Fig in S1 Materials). Funnel plot and Egger's meta-regression test also did not indicate the presence of publication bias (S5 Fig and S1 Table in S1 Materials).

## Risk factors of household transmission

The risk of transmission varies by the symptom status of the index case. Based on three studies with available data, household SAR of symptomatic index cases were significantly higher than asymptomatic and pre-symptomatic cases, with a relative risk (RR) of 3.23 (95% CI: 1.46, 7.14) (Fig 3). In all three studies, the household SAR of symptomatic index cases (20.0%; 95% CI: 11.4%, 28.6%) was higher than those of asymptomatic ones (4.7%; 95% CI: 1.1%, 8.3%) (Fig 4).

SAR from 14 studies showed that close contacts who were adults were more likely to be infected compared to children (< 18 years old), with a relative risk of 1.71 (95% CI: 1.35, 2.17)

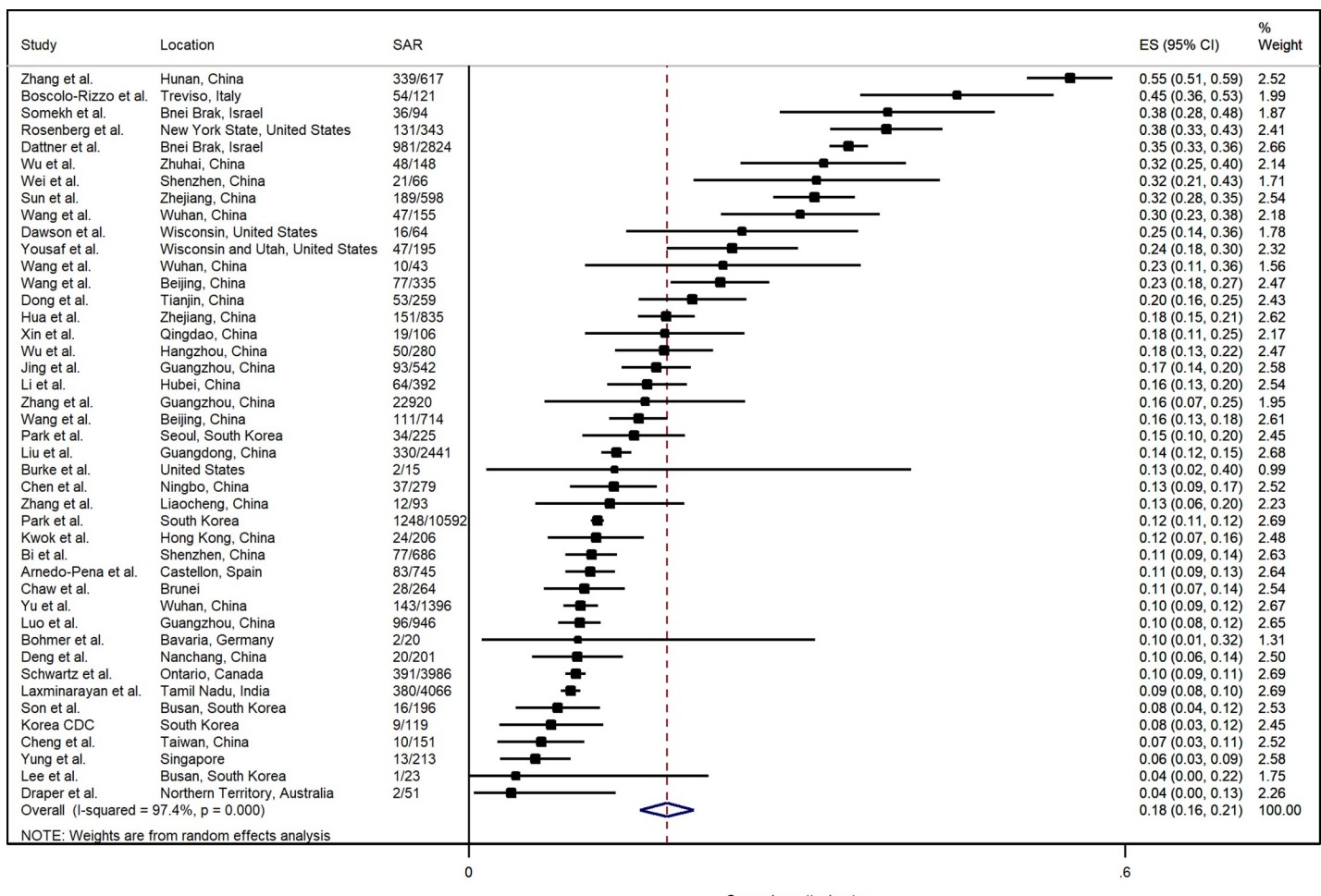

**Fig 2. Forest plot of household secondary attack rates (SAR).** ES is the estimated SAR, with 95% confidence intervals (CI). I-squared is the percentage of between-study heterogeneity that is attributable to variability in the true effect, rather than sampling variation.

(Fig 5). However, there was considerable heterogeneity among the included studies. In three studies, infection in adults was marginally lower than in children, but overall, the household SAR in adults (33.3%; 95% CI: 24.4%, 42.1%) was significantly higher than that in children (16.9%; 95% CI: 10.9%, 22.9%) (Fig 6).

Spouse relationship to index case from seven studies indicated a significantly higher risk of infection (RR: 2.39; 95% CI: 1.79, 3.19) compared to other household members (Fig 7). In all seven studies, the SAR to spouses (37.5%; 95% CI: 22.2%, 52.7%) was higher than to other household contacts (16.3%; 95% CI: 10.6%, 22.1%) (Fig 8). However, there was considerable heterogeneity among the included studies.

## Healthcare SAR

There are fewer SAR studies in non-household settings. We identified 18 studies that allowed direct estimation of the SAR in healthcare settings where transmission was determined to arise from an infected patient (Table 2). Nine of the studies covered multiple settings while the other nine studies focused solely on transmission in healthcare settings.

Sixteen studies were published articles (two in Chinese language) and two were pre-prints. Nine studies were in China, four in the United States, and the others were in Germany, India,

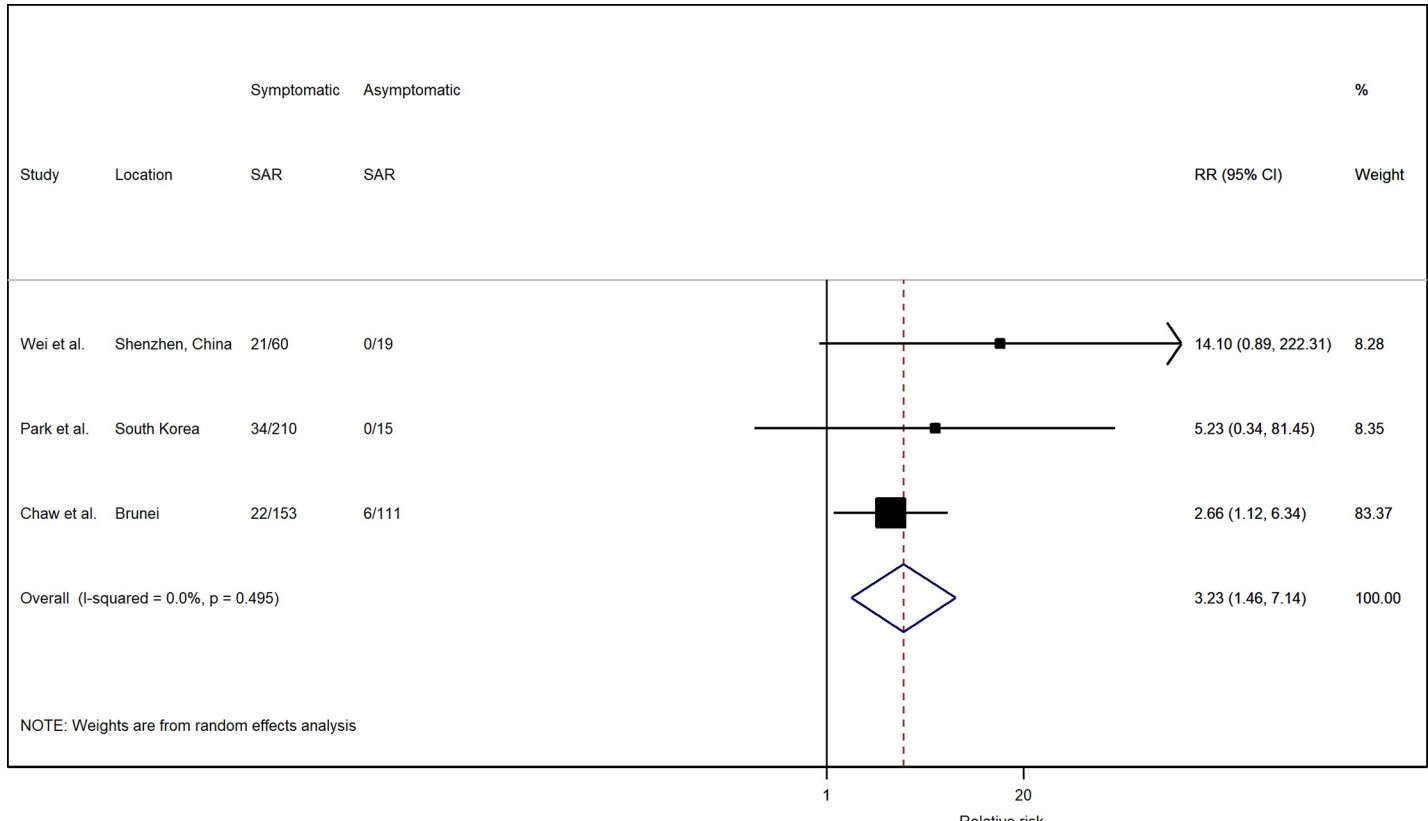

**Fig 3. Forest plot of household transmission risk by symptom status of index case.** RR is the estimated risk ratio, with 95% confidence intervals (CI). I-squared is the percentage of between-study heterogeneity that is attributable to variability in the true effect, rather than sampling variation.

Japan, Singapore, and Switzerland. All close contacts were tested regardless of symptoms except for four studies where testing was done only on symptomatic contacts. There was minor variation in the definition of healthcare contacts; most included healthcare workers and patients that were exposed to the index case, although a few studies were more specific in indicating close contact as those without personal protective equipment (PPE) or within a certain distance from the index case.

Fig 9 summarizes the estimated SARs. The pooled healthcare SAR was 0.7% (95% CI: 0.4%, 1.0%). Heterogeneity was not significant (p = 0.690). The SAR in healthcare settings in most studies was generally low (< 2%), except for a study in Wuhan that indicated 2 of 5 (40%) healthcare personnel were infected [37]. A study in California that tested symptomatic contacts only [68] had a relatively high healthcare SAR (7.0%), but overall there was no significant difference according to testing protocols (S6 Fig in S1 Materials).

## SAR in other non-household settings

We found 17 studies that allowed estimation of SAR in settings or by contact type other than household and healthcare: relatives outside the household; meal; travel; social; workplace; school; religious gathering; business meeting; choir; and chalet (Table 3). Due to the limited number of studies in each of these settings, unclear or imprecise definitions of close contacts, and the large variation in SAR across the settings, we did not estimate a pooled SAR. Instead, we reported the SAR to highlight potential high-risk settings.

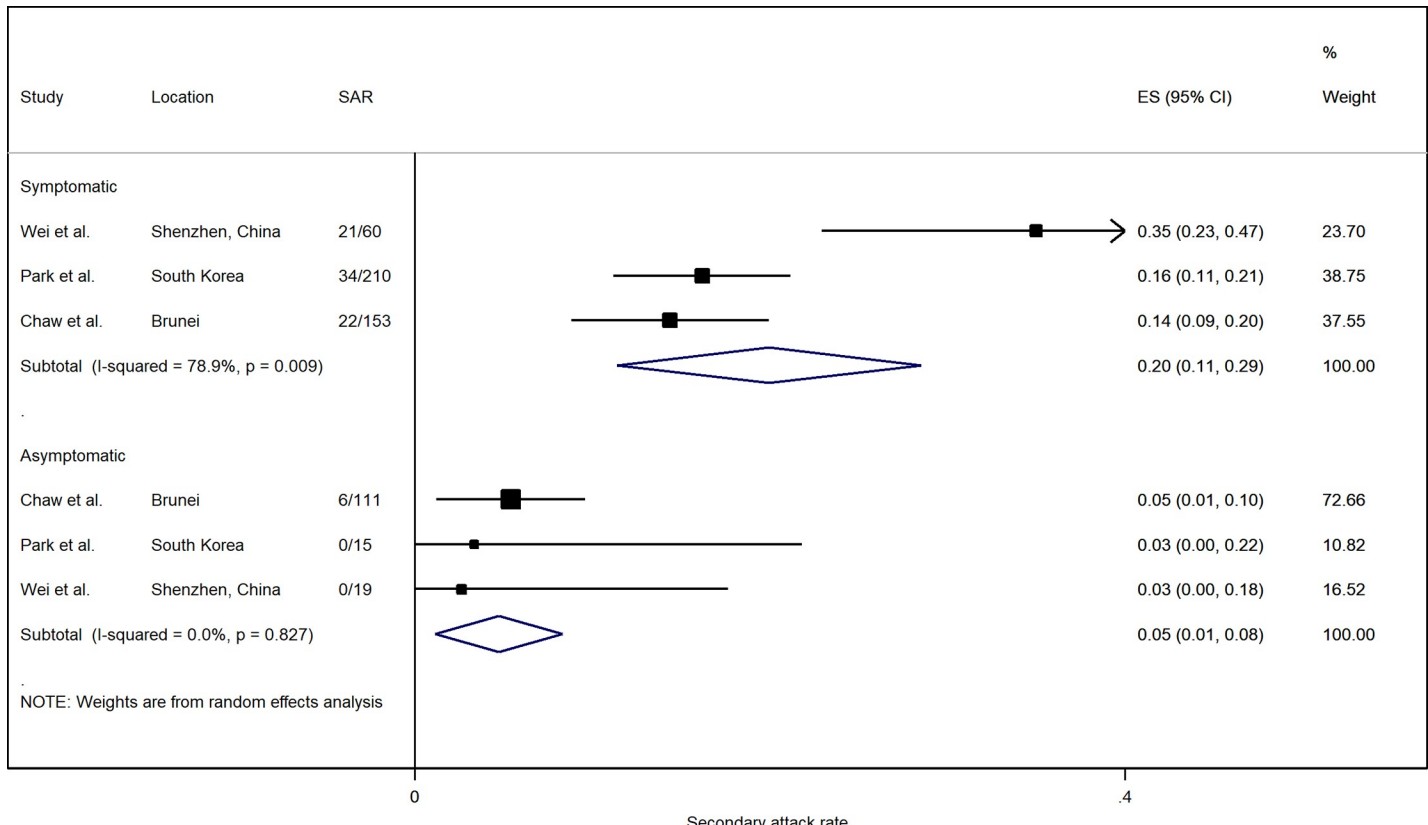

**Fig 4. Forest plot of household secondary attack rates (SAR) by symptom status of index case.** ES is the estimated SAR, with 95% confidence intervals (CI). I-squared is the percentage of between-study heterogeneity that is attributable to variability in the true effect, rather than sampling variation.

High SARs were observed in a meeting (84.6%), a chalet (73.3%), and at choirs (70.4%, 53.3%). In other settings, relatively high SARs were reported in eating (38.8%, 28.6%) and traveling (80.8%, 46.6%) with a case, as well as a study evaluating a religious event (14.8%). SARs were much lower in encounters with relatives (3.5% to 6.6%), social contacts (0.9% to 2.2%), and at workplace or school (0% to 5.3%).

## Discussion

### Summary of key findings

We estimated household SAR at 18.1% (95% CI: 15.7%, 20.6%), with significant heterogeneity across studies ranging from 3.9% to 54.9%. Symptomatic persons in households had a significantly higher risk of infecting others compared to asymptomatic ones (RR: 3.23; 95% CI: 1.46, 7.14). Adults in households had a significantly higher risk of infection relative to children (RR: 1.71; 95% CI: 1.35, 2.17). Spouses of index cases were more likely to be infected when compared to other household contacts (RR: 2.39; 95% CI: 1.79, 3.19). In healthcare settings, SAR was estimated at 0.6% (95% CI: 0.4%, 0.9%).

### Secondary attack rate

We used SAR across various settings as a measure of viral transmissibility. While a number of studies have estimated the basic reproductive number (R0) at 2–4, [77–80] in isolation it is a

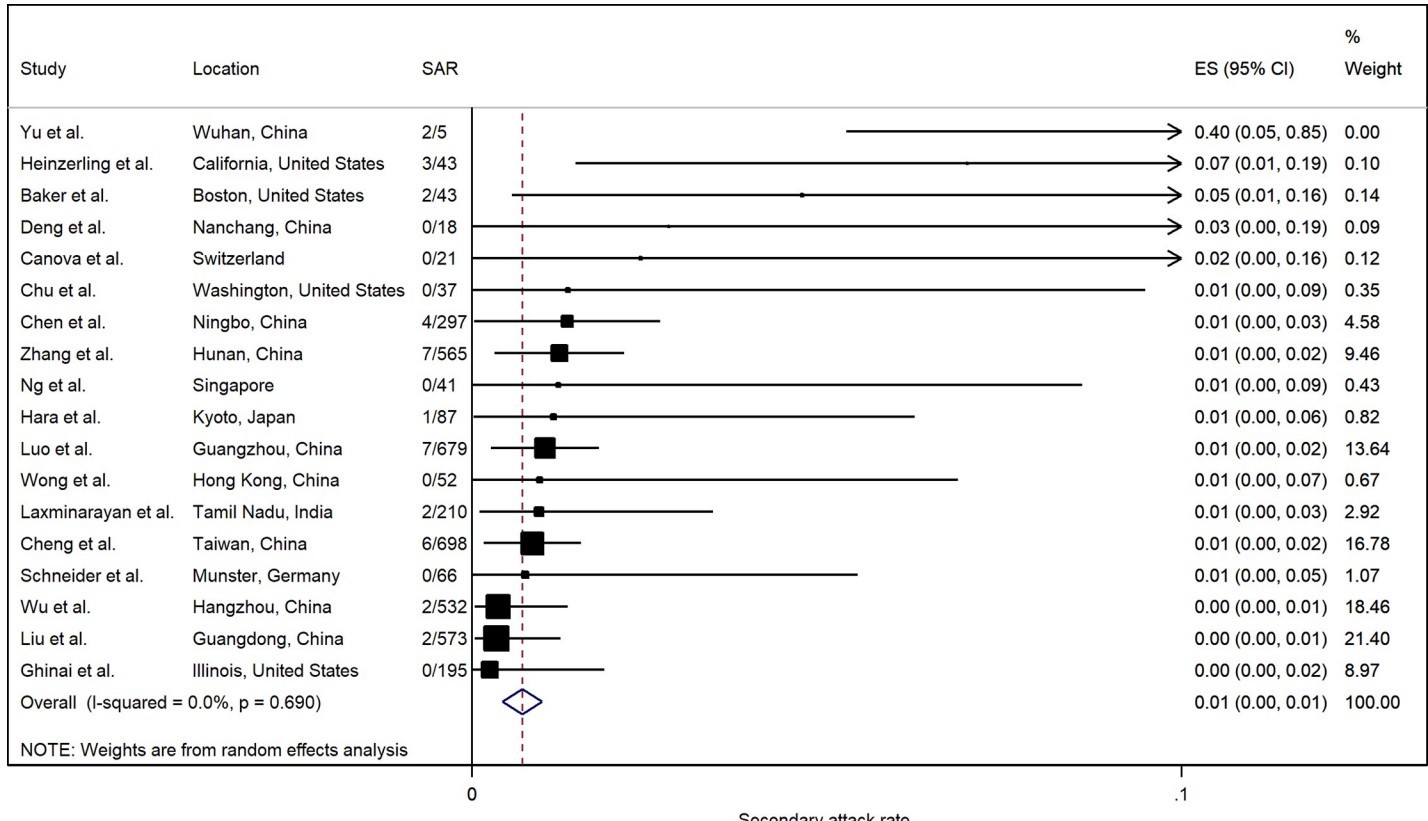

**Fig 5. Forest plot of household transmission risk by adult and children close contact.** RR is the estimated risk ratio, with 95% confidence intervals (CI). I-squared is the percentage of between-study heterogeneity that is attributable to variability in the true effect, rather than sampling variation.

suboptimal gauge of infectious disease dynamics as it does not account for variability in specific situations and settings [81, 82].

Significant heterogeneity in SAR across different settings is unsurprising given that SAR depends not only on the causative agent but also on socio-demographic, environmental, and behavioral factors in study populations [83]. Variation in methods for case ascertainment and subsequent detection of infected cases among contacts likely contributed to the heterogeneity across studies.

Household SAR was estimated at 18.1%. Reports suggest that familial transmission account for the majority of transmissions [36, 84]. The household is thought to be a fundamental unit of SARS-CoV-2 transmission because of the high frequency and intensity of contacts that occur between family members, and because transmission has continued in places with movement restriction [44]. We found that household SAR was higher than the upper range of estimates of the household SAR for the 2009 H1N1 pandemic influenza (5–15%) [85–87], and also higher than that observed for both SARS (5–10%) [88–90] and MERS (4–5%) [91, 92]. This suggests relatively higher SARS-CoV-2 transmissibility in the household setting, when compared to that of H1N1 and MERS viruses. SARS-CoV-2 also has a higher R0 when compared to MERS-CoV and SARS-CoV-1 [93]. This finding highlights the necessity of swift case isolation, immediate tracing, and quarantine of household contacts [94].

The highest household SARs were observed in mainland China, Israel, Italy, and the United States—countries with sustained outbreaks—whereas SARs were generally lower in countries and areas that have done relatively well in outbreak control, such as Brunei, Hong Kong, South

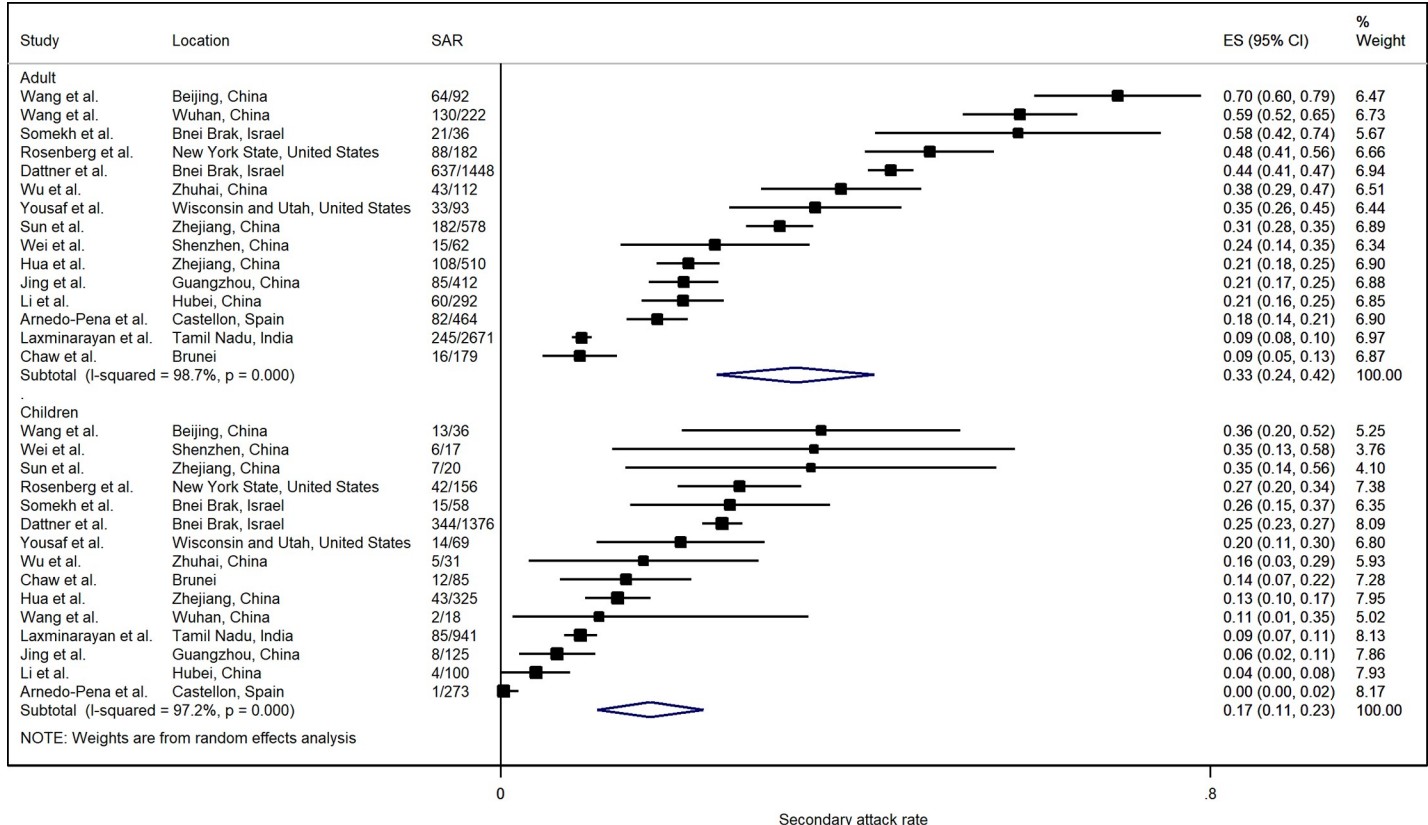

**Fig 6. Forest plot of household secondary attack rates (SAR) by adult and children close contact.** ES is the estimated SAR, with 95% confidence intervals (CI). I-squared is the percentage of between-study heterogeneity that is attributable to variability in the true effect, rather than sampling variation.

Korea, and Taiwan. Outside sources of infection are likely to be higher in countries with sustained community transmission, and as such without accounting for these, the household SARs are likely to be overestimated. Nonetheless, the potential for high transmission in households is clearly evident.

Healthcare workers who provide care to hospitalized patients could be at high risk of infection, particularly those without adequate PPE due to delayed diagnosis of COVID-19. We quantified this risk and found that SARs in healthcare settings in most studies were low ($< 2\%$). An exception is a study in Wuhan, which reported that 2 out of 5 (40%) medical personnel were infected [37]. The authors attributed the high SAR to inadequate acknowledgment of pathogens, misclassification of patients with COVID-19 as ordinary fever cases, and shortage of PPE during the early stage (late December 2019 to early January 2020) when the outbreak was still not well understood.

The generally low SAR in non-household settings may mask variation between setting types. Some studies reported significantly higher SAR in mass gatherings and other enclosed settings with potential for prolonged physical contact, such as at a meeting in Germany (84.6%) [75], a ski chalet in France (73.3%) [71], at a choir in France (70.4%) [72], during meals in China (38.8%) [40], and during travel in India (80.8%) [47]. In contrast, SAR in workplace, school, and social settings ranged between 0–5%, suggesting a gradation of risk outside the household.

Our meta-analyses excluded studies that solely reported attack rates (AR) without identification of an index case and their transmission generations within the cluster. However, such

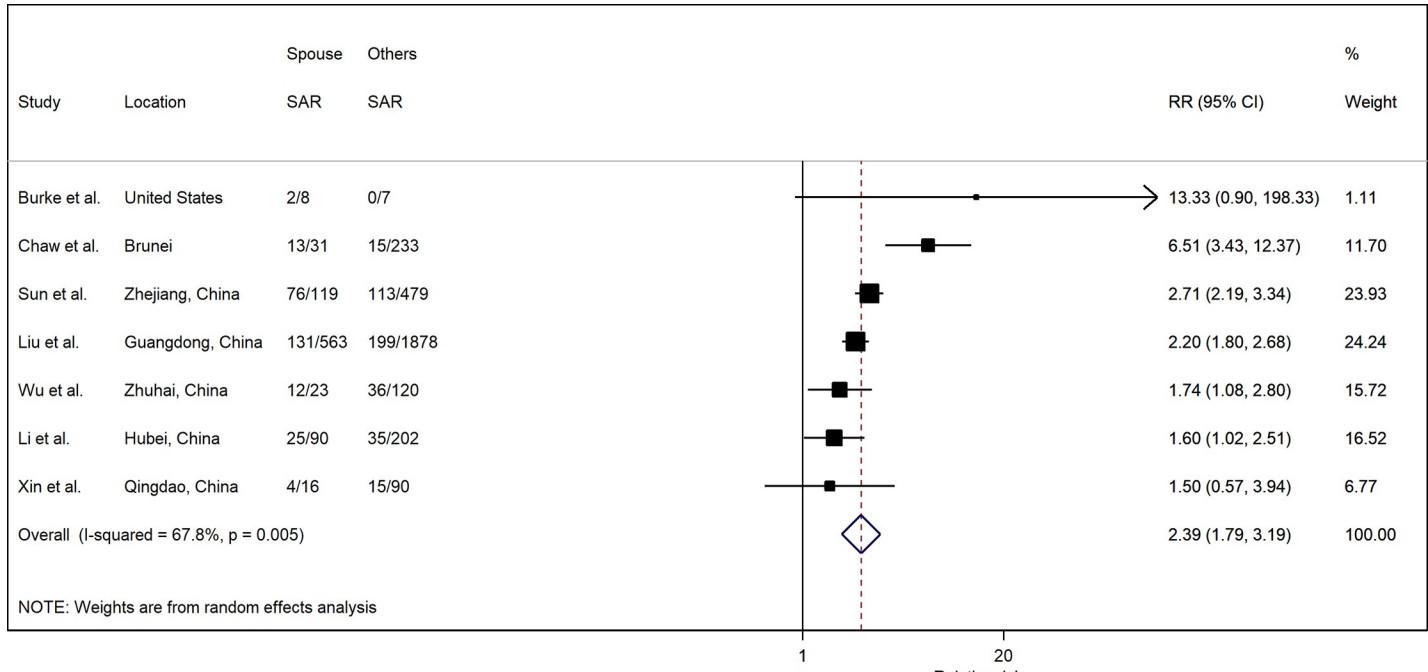

**Fig 7. Forest plot of household transmission risk by relationship to index case.** RR is the estimated risk ratio, with 95% confidence intervals (CI). I-squared is the percentage of between-study heterogeneity that is attributable to variability in the true effect, rather than sampling variation.

studies may be important in understanding the role of super-spreading events (SSEs) in driving SARS-CoV-2 transmission [82]. Specific settings where high ARs (> 20%) have been observed were in a correctional and detention facility in Louisiana (72.4%) [95], nursing homes in California (70.3%) [96] and the United Kingdom (40.3%) [97], in cruise ships (59%) [98], a call centre in South Korea (43.5%) [54], a church in Arkansas (38%) [99], among college students during a spring break trip in Mexico (32.8%) [100], a homeless shelter in Boston (36%) [101], a fitness dance class in South Korea (26.3%) [102], and a wedding in Jordan (21.7%) [103] (S2 Table in S1 Materials). High ARs have also been reported in healthcare settings in Mexico City (31.9%) [104] and the United Kingdom (27.7%) [105].

Reflecting on the high SAR in households and high AR in numerous non-household settings, we suggest that several common environmental factors could potentially account for the rapid person-to-person transmission observed: closed environments, population density, and shared eating environments. This is supported by environmental sampling studies [106] and from ecological observations on the declining incidence of COVID-19 cases in areas with restrictions placed on indoor mass gatherings [107].

There are implications for mass gatherings, particularly as countries begin to relax physical distancing measures. Non-household residential settings such as long-term care facilities, dormitories, and detention facilities pose specific challenges where additional prevention measures merit consideration, including staff screening, enhanced testing, and strict visitor policies [108].

Certainly, across all settings, the longer the duration and the greater the degree of physical contact with an index case, the higher the risk of transmission. However, we find that the risk model for transmission of SARS-CoV-2 is nuanced—while the highest risk of transmission is in crowded and enclosed settings, casual social interaction in some public settings have a lower risk. In addition, as the pandemic progresses and concern with physical distancing measures

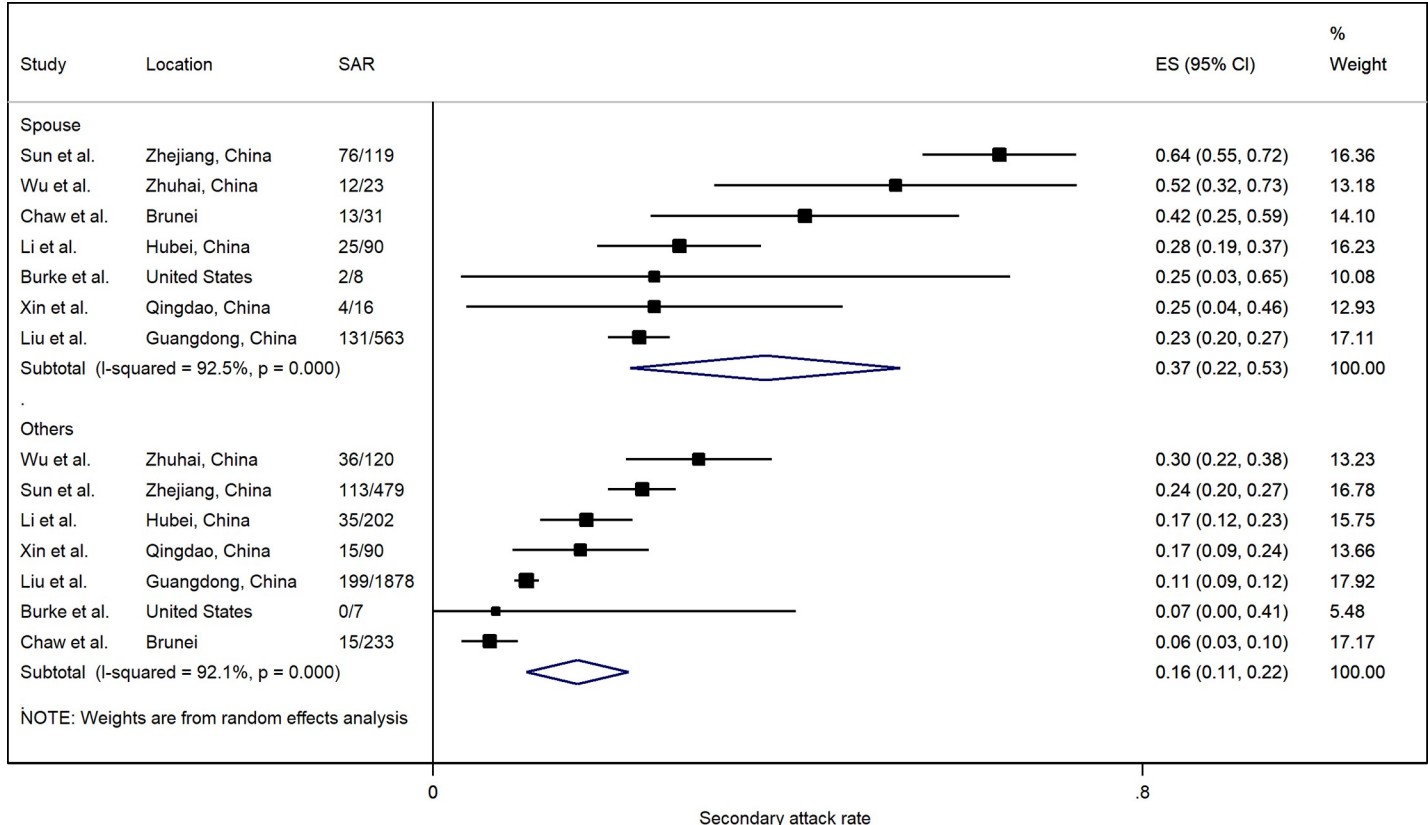

**Fig 8. Forest plot of household secondary attack rates (SAR) by relationship to index case.** ES is the estimated SAR, with 95% confidence intervals (CI). I-squared is the percentage of between-study heterogeneity that is attributable to variability in the true effect, rather than sampling variation.

(so-called "quarantine fatigue") gain momentum [109], public communications surrounding these measures should convey this continuum of risk based on the transmission dynamics across different settings, supporting sustainable longer-term behavior changes.

## SARS-CoV-2 transmission in children

For many infectious diseases, such as seasonal and pandemic influenza, children are known be drivers of transmission within households and communities [110]. Case series data on SARS--CoV-2 suggests that children are less likely to be affected than adults. A national analysis of the first 72,314 cases in China reported only 2.1% of all cases were children aged 0–19 years old [111]. Other population-wide studies show similarly low proportions [56, 112, 113].

To better understand their relative susceptibility to infection, we compared the SAR between adults and children and found that adults were at 1.7 times higher risk of infection than children. The lower rate of susceptibility in children could be explained by differences in symptomatic infection rates and subsequent issues with case ascertainment [114].

The literature surrounding infectivity in children was scarce. In household transmission studies, children were usually identified through contact tracing of adult cases, although a number of case reports documented transmission from children to adults [115]. There is also insufficient knowledge on transmissibility of SARS-CoV-2 from children to other children. In addition, age may be important to determine dynamics of interactions among children but inadequate data hampered our efforts at risk stratification by age.

**Table 2. Description of studies included in the review and analysis of healthcare secondary attack rate (SAR).**

| Study, location | Description of study | Definition of close contact | Healthcare SAR (%) | No. of index cases | Additional comments | Quality score |
|---|---|---|---|---|---|---|
| Liu et al., Guangdong, China [21] | Retrospective study of different exposure settings | Healthcare workers exposed to case | 2/573 (0.3%) | 1361 | | 7 |
| Luo et al., Guangzhou, China [23] | Prospective study of different modes of contact | Medical staff who provide direct care, family members or others who have similar close contact with case, such as visiting or staying at the same hospital ward | 7/679 (1.0%) | 347 | | 7 |
| Wu et al., Hangzhou, China [25] | Retrospective study of different exposure settings | Healthcare provided or other patient | 2/532 (0.4%) | 144 | | 5 |
| Zhang et al., Hunan, China [27] | Retrospective study of different exposure settings | Diagnosed, treated, or nursed a case | 7/565 (1.2%) | 136 | | 6 |
| Deng et al., Nanchang, China [29] | Retrospective study of different exposure settings | Had medical services at the same time or shared wards | 0/18 (0%) | 27 | | 5 |
| Chen et al., Ningbo, China [30] | Prospective study of different exposure settings | Healthcare workers exposed to case | 4/297 (1.3%) | 187 | | 6 |
| Yu et al., Wuhan, China [37] | Retrospective study of different exposure settings | Doctors and patients exposed to case | 2/5 (40%) | 560 | | 5 |
| Wong et al., Hong Kong, China [62] | Retrospective study of healthcare setting | Patient or staff who stayed or worked in the same ward as the index patient | 0/52 (0%) | 1 | Only 52 of 120 contacts tested; the rest were asymptomatic | 5 |
| Cheng et al., Taiwan, China [42] | Prospective study of different exposure settings and different exposure time windows | Within 2m without appropriate PPE and without a minimal requirement of exposure time | 6/698 (0.9%) | 100 | Only close contacts with symptoms tested | 7 |
| Schneider et al., Munster, Germany [63] | Retrospective study of healthcare setting | Healthcare workers exposed to infected patient | 0/66 (0%) | 1 | | 5 |
| Laxminarayan et al., Tamil Nadu, India [47] | Retrospective study of different exposure settings | Healthcare workers exposed to case | 2/210 (1.0%) | 11 | | 7 |
| Hara et al., Kyoto, Japan [64] | Retrospective study of healthcare setting | Patients exposed to an infected healthcare worker | 1/87 (1.1%) | 1 | | 5 |
| Ng et al., Singapore [65] | Retrospective study of healthcare setting | Exposed to aerosol-generating procedures for at least 10 minutes at a distance of less than 2 meters from the infected patient | 0/41 (0%) | 1 | | 5 |
| Canova et al., Switzerland [66] | Analysis of contact investigation | Healthcare workers with unprotected contact with the case | 0/21 (0%) | 1 | | 6 |
| Baker et al., Boston, United States [67] | Retrospective study of healthcare setting | Provided care to infected patient | 2/44 (4.5%) | 1 | 7 healthcare workers not tested, and assumed negative | 5 |
| Heinzerling et al., California, United States [68] | Retrospective study of healthcare setting | Symptomatic healthcare workers exposed to infected patient | 3/43 (7.0%) | 1 | 121 healthcare workers exposed, but only those with symptoms tested | 5 |
| Ghinai et al., Illinois, United States [69] | Analysis of contact investigation | People who reported or were identified to have potential exposure on or after the date of symptom onset of the case | 0/195 (0%) | 1 | Only persons under investigation and selected asymptomatic healthcare personnel tested | 5 |
| Chu et al., Washington, United States [70] | Retrospective study of healthcare setting | Face-to-face interaction with infected patient without full personal protective equipment (PPE) | 0/37 (0%) | 1 | | 5 |

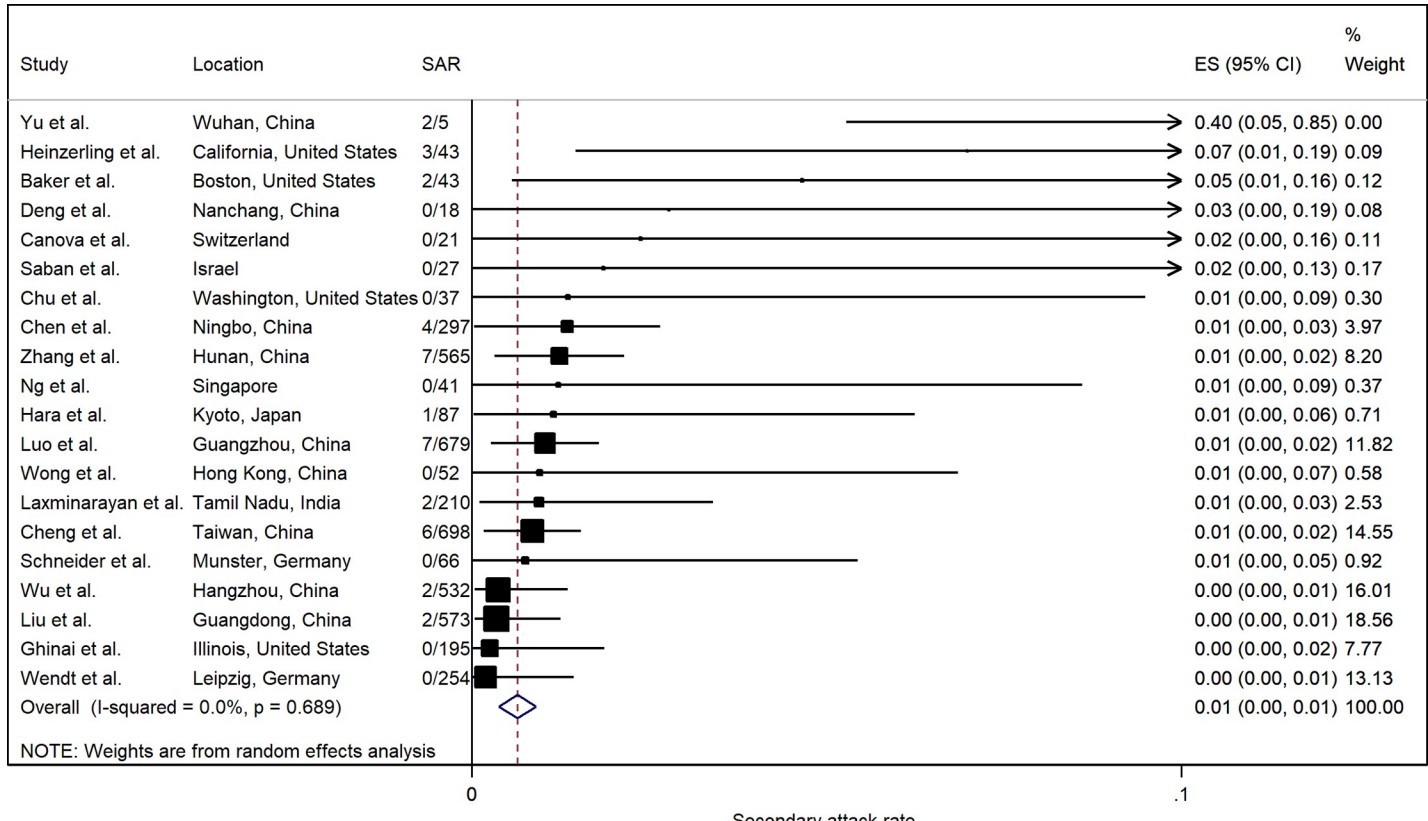

**Fig 9. Forest plot of secondary attack rates (SAR) in healthcare settings.** ES is the estimated SAR, with 95% confidence intervals (CI). I-squared is the percentage of between-study heterogeneity that is attributable to variability in the true effect, rather than sampling variation.

While there are important unknowns with respect to SARS-CoV-2 in children, these early findings may assist health authorities in determining proportionate thresholds for school closures in future waves of the pandemic.

## Strengths and limitations

Our analysis has important limitations. The studies selected were based on field investigation; variability was noted with respect to the study design, the number of individuals assessed, clinical definitions, the extent to which confirmatory laboratory tests were used, the methods of clinical data collection, and the duration of follow-up. Studies have different definitions of household and contacts and are subject to recall and observer bias [116]. Moreover, without accounting for outside sources of infection, setting-specific SARs are likely to be overestimated [83]. In fact, none of the reviewed studies addressed the composition of secondary vs. community infections when estimating the SAR or used viral sequencing to confirm homology between the strains infecting the index and secondary cases in the household.

All SAR studies were retrospective transmission studies based on contact tracing datasets where the index case determination or the direction of transmission may be uncertain, particularly as a substantial proportion of cases was asymptomatic or mild. An additional challenge concerns the timing of recruitment of cases and their contacts during the course of an epidemic. Studies conducted in early stages can provide timely SAR estimates; however, this may be influenced by behavioral factors and other non-pharmaceutical interventions (e.g. community quarantine) that could have altered over the course of the epidemic [83].

**Table 3. Studies of secondary attack rate (SAR) in settings outside household and healthcare.**

| Study | Location | Setting | SAR (%) |
|---|---|---|---|
| Danis et al. [71] | French Alps | Chalet | 11/15 (73.3%) |
| Charlotte [72] | France | Choir | 19/27 (70.4%) |
| Hamner et al. [73] | Washington, United States | Choir | 32/60 (53.3%) |
| Wu et al. [40] | Zhuhai, China | Meal | 40/103 (38.8%) |
| Shen et al. [74] | Zhejiang, China | Meal | 2/7 (28.6%) |
| Deng et al. [29] | Changsha, China | Meal | 17/160 (10.6%) |
| Bi et al. [32] | Shenzhen, China | Meal | 61/707 (8.6%) |
| Chen et al. [30] | Ningbo, China | Meal | 52/724 (7.2%) |
| Hijnen et al. [75] | Munich, Germany | Meeting | 11/13 (84.6%) |
| Cheng et al. [42] | Taiwan, China | Non-household family | 5/76 (6.6%) |
| Liu et al. [21] | Guangdong, China | Non-household family | 132/2266 (5.8%) |
| Chaw et al. [44] | Brunei | Non-household family | 5/144 (3.5%) |
| Chaw et al. [44] | Brunei | Religious | 8/54 (14.8%) |
| Wang et al. [20] | Beijing, China | Social | 75/3363 (2.2%) |
| Zhang et al. [24] | Guangzhou, China | Social | 1/66 (1.5%) |
| Liu et al. [21] | Guangdong, China | Social | 41/3344 (1.2%) |
| Chaw et al. [44] | Brunei | Social | 4/445 (0.9%) |
| Laxminarayan et al. [47] | Tamil Nadu, India | Travel | 63/78 (80.8%) |
| Wu et al. [40] | Zhuhai, China | Travel | 34/73 (46.6%) |
| Chen et al. [30] | Ningbo, China | Travel | 28/235 (11.9%) |
| Zhang et al. [24] | Hunan, China | Travel | 22/304 (7.2%) |
| Bi et al. [32] | Shenzhen, China | Travel | 18/318 (5.7%) |
| Draper et al. [43] | Northern Territory, Australia | Travel | 2/46 (4.3%) |
| Liu et al. [21] | Guangdong, China | Travel | 10/2778 (0.4%) |
| Luo et al. [23] | Guangzhou, China | Travel | 3/2358 (0.1%) |
| Deng et al. [29] | Changsha, China | Travel | 0/17 (0%) |
| Danis et al. [71] | French Alps | School | 0/112 (0%) |
| Heavey et al. [76] | Ireland | School | 0/1025 (0%) |
| Deng et al. [29] | Changsha, China | Workplace | 5/94 (5.3%) |
| Zhang et al. [24] | Guangzhou, China | Workplace | 0/119 (0%) |
| Chen et al. [30] | Ningbo, China | Workplace/school | 1/47 (2.1%) |
| Chaw et al. [44] | Brunei | Workplace/school | 6/848 (0.7%) |

The major strength of our study is that it comprehensively covers publicly available studies on SARS-CoV-2 transmission-related dynamics with regards to settings and associated risk factors, thus allowing a better understanding and identification of the key drivers of transmission.

## Conclusion

Our estimates of SAR across various settings demonstrate the challenges in controlling SARS-CoV-2 transmission. Overall, these findings suggest that aggressive contact-tracing strategies based on suspect cases may be appropriate early in an outbreak. However, as the outbreak progresses, control measures should transition to a combination of approaches that account for setting-specific transmission risk. Given the high SARs observed in households and other residential settings, physical distancing measures may need to cover entire communities such as dormitories, workplaces, or other institutional settings, while contact tracing should shift to

identifying hotspots of transmission and vulnerable populations. Where possible, confirmed cases should be isolated away from the household.

## Supporting information

**S1 Checklist. PRISMA 2009 checklist.**
(DOC)

**S1 Materials.**
(PDF)

## Author Contributions

**Conceptualization:** Wee Chian Koh, Justin Wong.

**Data curation:** Wee Chian Koh, Lin Naing, Muhammad Ali Rosledzana, Mohammad Fathi Alikhan, Justin Wong.

**Formal analysis:** Wee Chian Koh, Sirajul Adli Jamaludin, Faezah Amin, Asiah Omar, Alia Shazli.

**Methodology:** Wee Chian Koh, Lin Naing, Liling Chaw, Justin Wong.

**Supervision:** Justin Wong.

**Validation:** Justin Wong.

**Visualization:** Wee Chian Koh.

**Writing – original draft:** Wee Chian Koh, Justin Wong.

**Writing – review & editing:** Wee Chian Koh, Lin Naing, Liling Chaw, Muhammad Ali Rosledzana, Mohammad Fathi Alikhan, Sirajul Adli Jamaludin, Faezah Amin, Asiah Omar, Alia Shazli, Matthew Griffith, Roberta Pastore, Justin Wong.

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
