## [Decision Letter · Decision Letter 0]

6 Jul 2020

PONE-D-20-15727

What do we know about SARS-CoV-2 transmission? A systematic review and meta-analysis of the secondary attack rate, serial interval, and asymptomatic infection.

PLOS ONE

Dear Dr. Wong,

Thank you for submitting your manuscript to PLOS ONE. After careful consideration, we feel that it has merit but does not fully meet PLOS ONE’s publication criteria as it currently stands. Therefore, we invite you to submit a revised version of the manuscript that addresses the points raised during the review process.

The primary issue raised during the review process is that there is sufficient heterogeneity in the included studies with respect to settings and methods such that a pooled estimate is not appropriate and in fact may be misleading from a policy standpoint. Therefore, we recommend presenting stratified estimates by groups that are more homogeneous vs. a single pooled estimate.

In addition to the recommendations from reviewers, I have the following suggestions:

1. Given the relevance of asymptomatic cases and asymptomatic transmission, I would also stratify SAR estimates based on whether studies tested all contacts vs. only symptomatic contacts. 

2. Similarly, it would be helpful to include the total number of "source" cases in each study, given that this is used later in the estimates of transmission from symptomatic vs asymptomatic cases. This is an important aspect of the paper but the methods are not as well described. The finding of difference in risk of transmission from symptomatic vs asymptomatic cases would be of considerable interest and should be expanded both in methods and discussion of the findings. 

We look forward to receiving your revised manuscript.

Kind regards,

Surbhi Leekha

Academic Editor

PLOS ONE

Journal Requirements:

Reviewers' comments:

Reviewer's Responses to Questions

**Comments to the Author**

1. Is the manuscript technically sound, and do the data support the conclusions?

Reviewer #1: Partly

Reviewer #2: Yes

2. Has the statistical analysis been performed appropriately and rigorously? 

Reviewer #1: No

Reviewer #2: Yes

3. Have the authors made all data underlying the findings in their manuscript fully available?

Reviewer #1: Yes

Reviewer #2: Yes

4. Is the manuscript presented in an intelligible fashion and written in standard English?

Reviewer #1: Yes

Reviewer #2: Yes

5. Review Comments to the Author

Reviewer #1: In general, too much information is being forced into one manuscript leading to such high heterogeneity that no real conclusions can be made. I recommend that this be separated into multiple manuscripts in order to find homogeneous populations that can lead to generalizations. For example, it is obvious that SAR would be different in healthcare setting vs. travel vs. people sharing a meal. Lumping these together dilutes the real risk in high-risk settings like sharing a meal. The authors make this same argument in the discussion section, but do not use their recommended nuanced approach in the methods. In this era when COVID guidelines are being written, a “non-household” SAR of 4% may lead to governments stating that it is fine to travel and share meals. The authors need to be very confident in results like this (which they cannot be since there is high heterogeneity) to make a conclusion like that. The conclusions overreach the data and are confusing. We live in a setting where journalists will take conclusions from published abstracts about COVID-19 and use them as headlines in international newspapers. Thus, we need to be extra vigilant that the conclusions match the data.

My recommended edits are as follows:

1) The forest plot in Figure 1 has so much heterogeneity that it is hard to make any conclusions about these studies.

a) You should NOT pool “non-household” studies together but rather do subset analyses of more homogenous groups like healthcare settings.

b) Since you state that studies used different definitions of household contacts, and your household contact SAR analysis was highly heterogeneous, it would make sense to do a stratified analysis by definition of household contact.

c) Also consider stratifying the household SAR analysis into subsets by spouse, child or “other” household members, not as a RR but as a SAR for each group.

2) Similarly, the forest plot in Figure 3 has so much heterogeneity that it is hard to make any conclusions about these studies. These should also be broken into more homogeneous subsets.

3) In every summary (e.g., discussion, abstract) you need to state that the studies were heterogeneous. The included studies are much too different from one another to make any sort of generalizable summary statement. The results section of the abstract should also include the full range of values, not just the 95% confidence intervals, because there is so much heterogeneity.

4) This meta-analysis is missing a key component of the PRISMA checklist: Risk of bias in individual studies. Please rate the risk of bias from each study and comment on how the studies with moderate-to-high risk of bias make it difficult for conclusions to be made. There are many risk of bias tools available such as the Institute of Health Economics (IHE) Quality Apprasial of Case Series Studies Checklist Institute of Health Economics, 2014. 2019, at http://www.ihe.ca/research-programs/rmd/cssqac/cssqac-about.)

5) Many times, I could not determine whether a subset analysis was done or if the authors just averaged results together without taking into account study weight nor calculating confidence intervals. For example, the asymptomatic proportion of pregnant women needs to be calculated as a meta-analysis with a confidence interval, not just an average of studies.

6) The confidence interval calculations for SAR do not make biologic sense. You can’t have a negative SAR or a SAR more than 100%.

7) Potentially missing multiple secondary attack rate studies. There are many analyses of contact investigations not included such as:

Heinzerling et al: https://www.cdc.gov/mmwr/volumes/69/wr/mm6915e5.htm?s_cid=mm6915e5_w

Pung et al: https://www-sciencedirect-com.proxy.lib.uiowa.edu/science/article/pii/S0140673620305286?via%3Dihub

8) The fitted values in the scatter plots of asymptomatic proportion in the supplement are not very believable since the data points are few and scattered. I would recommend doing a stratified analysis with categorize ages rather than treating them as continuous.

9) The conclusions overreach the data and are confusing. It doesn’t make sense to say that 10 symptomatic persons living with 100 contacts would result in 15 cases because it would be incredibly rare to have 100 household contacts. Either use a more realistic example or do not use an example like this at all. Similarly, the statement “effective quarantine needs to be implemented within 3 days” does not seem fact based. Where did the number 3 come from? Also, the statements about “targeted physical distancing strategies focusing on high-density enclosed settings” and statements about dormitories and workplaces are not supported by your data because you lump together all non-household interactions.

10) I am confused about whether non-English studies were excluded. It is poor practice to exclude studies based on language, especially when an author is from WHO and has access to scientists who speak many languages.

MINOR

1) On page 9, >10 is used when I think you mean <10.

2) Need to say, “publication bias” or “reporting bias” not “small study effect.”

3) Figure 2 needs to be plotted on a log scale, so the confidence intervals are symmetrical.

4) I would also recommend performing stratified analysis by publication status. Are the results different among studies that have not yet been peer reviewed compared with peer reviewed studies?

5) The discussion section needs to include more limitations. For example, the asymptomatic proportion may be overestimated for the reasons that you state in the results section, but also may be underestimated because people tested may not be representative of the general population.

6) Confidence intervals that include 1.00 mean that there is not a statistically significant difference at the p<0.05 level. Please re-word all of your statements about the RR=1.40 comparing adults vs. children since this confidence interval includes 1.

Reviewer #2: Overall, the manuscript is clear and well written, and is a systematic look at several important parameters, along with a discussion of their implications. I have a few suggestions to improve the manuscript as it currently stands:

SAR definition: "...dividing the number of exposed close contacts who tested

positive (numerator) by the total number of close contacts of the index case (denominator)." Should this not be exposed close contacts for both the numerator and denominator for consistency? Those not at risk (i.e. not exposed) should not be in the denominator.

Figure S2b: To me, this funnel plot is extremely alarming. At the moment, this is sort of discussed as a side note, but I think it's hard to argue that there's anything other than a pretty systematic bias in terms of the currently published studies on this particular parameter. Given that this is also the parameter that impacts a good deal of the discussion and the implications for policy and interventions, the authors need to spend some more time on this.

Finally, as with all reviews during an emerging epidemic, I suggest a search of the literature between May and now and a re-analysis to ensure that the results are timely as of publication.

6. PLOS authors have the option to publish the peer review history of their article (what does this mean?). If published, this will include your full peer review and any attached files.

Reviewer #1: **Yes: **Marin Schweizer

Reviewer #2: No

---

## [Author Response · Author response to Decision Letter 0]

11 Aug 2020

Please see attached document "Response to Reviewers" for a point by point response

---

## [Decision Letter · Decision Letter 1]

7 Sep 2020

PONE-D-20-15727R1

What do we know about SARS-CoV-2 transmission? A systematic review and meta-analysis of the secondary attack rate and associated risk factors.

PLOS ONE

Dear Dr. Wong,

Thank you for submitting your manuscript to PLOS ONE. After careful consideration, we feel that it has merit but does not fully meet PLOS ONE’s publication criteria as it currently stands. Therefore, we invite you to submit a revised version of the manuscript that addresses the points raised during the review process.

Overall, the manuscript is quite improved and we thank you for incorporating previous suggestions. However, there are a few additional comments provided by a reviewer and editor below.

We look forward to receiving your revised manuscript.

Kind regards,

Surbhi Leekha

Academic Editor

PLOS ONE

Additional Editor Comments (if provided):

1. Can the authors explain the column header “number of index cases” in more detail in the text, as well add a footnote to the table, particularly as in some cases, the number of index cases is larger than the number of contacts. How was the symptom status of the index case determined when there were multiple index cases?

2. Most of the healthcare settings included look at transmission from infected case patients, however a couple of them describe transmission from infected healthcare workers. I would describe those separately, as the latter falls under healthcare workplace “community” transmission, with different mitigation and control measures, whereas the former has implications for risk to healthcare workers during patient care delivery with different risks (e.g., medical procedures) and mitigation via PPE and other protocols.

Reviewers' comments:

Reviewer's Responses to Questions

**Comments to the Author**

1. If the authors have adequately addressed your comments raised in a previous round of review and you feel that this manuscript is now acceptable for publication, you may indicate that here to bypass the “Comments to the Author” section, enter your conflict of interest statement in the “Confidential to Editor” section, and submit your "Accept" recommendation.

Reviewer #1: (No Response)

2. Is the manuscript technically sound, and do the data support the conclusions?

Reviewer #1: Partly

3. Has the statistical analysis been performed appropriately and rigorously? 

Reviewer #1: Yes

4. Have the authors made all data underlying the findings in their manuscript fully available?

Reviewer #1: Yes

5. Is the manuscript presented in an intelligible fashion and written in standard English?

Reviewer #1: Yes

6. Review Comments to the Author

Reviewer #1: This revised version is greatly improved. However, some changes still need to be made so that the statements made do not overreach the data.

1. The meta-analyses shown in Figures 5 and 8 have considerable heterogeneity (the Cochrane Handbook defines considerable heterogeneity as I-squared >75%). When these results are described on pages 9 and 10, sentences need to be added that say “However, there was considerable heterogeneity among the included studies.”

2. The discussion section is overreaching the data. This study did not assess community lockdowns and restricting social movement. Thus, the discussion should not make conclusions about them on page 15 and the final sentence on page 16. It is unknown what the SAR would have been if communities did not restrict social movement when they did.

Minor comment page 13: I think there is a typo here and it should say “with a STUDY as well evaluating a religious event” as there were 8 cases at the religious event not one.

Minor comments page 15: 1) I don’t think anxiety is the right word when discussing quarantine fatigue; 2) should be “to better understand their relative susceptibility TO infection” not “of infection.”

7. PLOS authors have the option to publish the peer review history of their article (what does this mean?). If published, this will include your full peer review and any attached files.

Reviewer #1: No

---

## [Author Response · Author response to Decision Letter 1]

21 Sep 2020

REVIEWER COMMENTS

REVIEWER: The meta-analyses shown in Figures 5 and 8 have considerable heterogeneity (the Cochrane Handbook defines considerable heterogeneity as I-squared >75%). When these results are described on pages 9 and 10, sentences need to be added that say “However, there was considerable heterogeneity among the included studies.”

RESPONSE: We have added these sentences in our description of the results on p.9 and p.10.

REVIEWER: The discussion section is overreaching the data. This study did not assess community lockdowns and restricting social movement. Thus, the discussion should not make conclusions about them on page 15 and the final sentence on page 16. It is unknown what the SAR would have been if communities did not restrict social movement when they did. 

RESPONSE: We have removed the sentences alluding the community lockdowns and movement restrictions on p.15 and p.16.

REVIEWER: Minor comment page 13: I think there is a typo here and it should say “with a STUDY as well evaluating a religious event” as there were 8 cases at the religious event not one. 

RESPONSE: Thank you for this comment, we have amended this.

REVIEWER: Minor comments page 15: 1) I don’t think anxiety is the right word when discussing quarantine fatigue; 2) should be “to better understand their relative susceptibility TO infection” not “of infection.” 

RESPONSE: Thank you for this comment, we have rephrased this sentence on p.15, and amended (2)

ADDITIONAL EDITOR COMMENTS 

EDITOR: Can the authors explain the column header “number of index cases” in more detail in the text, as well add a footnote to the table, particularly as in some cases, the number of index cases is larger than the number of contacts. How was the symptom status of the index case determined when there were multiple index cases?

RESPONSE: We have included a description for this on p.6-7 of the manuscript. In general, index cases were confirmed positive cases identified or suspected to have been first exposed to the SARS-CoV-2 virus within the household, generally based on the timing of symptom onset and epidemiological link. Some of the studies did not specify the number of index cases, which we have marked as “n/a” in Table 1. As we explained in the “Strengths and limitations” section, the studies are based on contact tracing where the index case determination or the direction of transmission may be uncertain, particularly as a substantial proportion of cases was asymptomatic or mild.

EDITOR: Most of the healthcare settings included look at transmission from infected case patients, however a couple of them describe transmission from infected healthcare workers. I would describe those separately, as the latter falls under healthcare workplace “community” transmission, with different mitigation and control measures, whereas the former has implications for risk to healthcare workers during patient care delivery with different risks (e.g., medical procedures) and mitigation via PPE and other protocols.

RESPONSE: We take note of this comment and agree that there are likely to be differences in the transmission dynamics from an infected patient vs. a colleague. The former (18 studies) are what are more commonly perceived as true healthcare delivery settings type transmission – with, as noted, important implications for healthcare delivery risks, and the later (2 studies) can also be thought of as a workplace-type setting. Given this distinction, we have removed the 2 studies that describe transmission from infected healthcare workers from the pooled analysis.

---

## [Editor Report · Decision Letter 2]

23 Sep 2020

What do we know about SARS-CoV-2 transmission? A systematic review and meta-analysis of the secondary attack rate and associated risk factors.

PONE-D-20-15727R2

Dear Dr. Wong,

We’re pleased to inform you that your manuscript has been judged scientifically suitable for publication and will be formally accepted for publication once it meets all outstanding technical requirements.

Kind regards,

Surbhi Leekha

Academic Editor

PLOS ONE
---

## [Editor Report · Acceptance letter]

30 Sep 2020

PONE-D-20-15727R2 

What do we know about SARS-CoV-2 transmission? A systematic review and meta-analysis of the secondary attack rate and associated risk factors 

Dear Dr. Wong:

I'm pleased to inform you that your manuscript has been deemed suitable for publication in PLOS ONE. Congratulations! Your manuscript is now with our production department. 

Kind regards, 

on behalf of

Dr. Surbhi Leekha 

Academic Editor

PLOS ONE